# Flexible employment policies, temporal control and health promoting practices: A qualitative study in two Australian worksites

**Jane Dixon[1]***, **Cathy Banwell** **[1]**, **Lyndall Strazdins[1]**, **Lara Corr[2]**, **John Burgess[3]**

**1** Research School of Population Health, Australian National University, Canberra, Australian Capital Territory, Australia, **2** School of Arts, Social Sciences and Humanities, Faculty of Health, Arts and Design, Swinburne University of Technology, Melbourne, Victoria, Australia, **3** School of Management, Royal Melbourne University of Technology University, Melbourne, Victoria, Australia

ⓔ These authors contributed equally to this work.
* Jane.Dixon@anu.edu.au

**Data Availability Statement:** Data are available upon request due to legal and ethical restrictions for confidential data imposed by the Australian Government's Code for the Responsible Conduct

## Abstract

For four decades, theories of job demand-control have proposed that higher occupational status groups have lower health risks due to the stress accompanying jobs featuring high demands but high control. This research examines whether Flexible Work Arrangements (FWAs) can improve the health prospects of a range of workers by giving greater control over work time arrangements. Our setting is Australia, where FWAs were introduced in 2009. In line with these early studies alongside studies of work-life balance, we expected to observe that workers with access to control over daily work times could better control the activities outside of work that influence chronic disease. Using a practice sociology approach, we compared the accounts of twenty-eight workers in blue and white collar industries with differing degrees of work time flexibility. The findings do not contradict early theories describing occupational differences of job demand-control dynamics and their relationship to health risks. However, this study suggests that a) time demands and strains have increased for a broad sweep of workers since the 1980s, b) the greater control of higher occupational status groups has been eroded by the high performance movement, which has attracted less scrutiny than FWAs, and c) more workers are forced to adapt their daily lives, including their approach to health, to accommodate their job demands. Job insecurity further impedes preventative health practices adoption. What might appear to be worker-controlled flexibility can—under the pressures of job insecurity and performance expectations without time limits—transform into health-eroding unpredictability. The answer however is not greater flexibility in the absence of limits on the well-documented precursors of work stress: long hours, job insecurity and intensity-related exhaustion. While there have been welcome developments in job demand-control-health conceptualizations, they typically ignore the out-of-work temporal demands that workers face and which compound on-the-job demands. Redesign of the temporalities of working life within worksites need to be accompanied by society-level policies which address caring responsibilities, gender equality as well as broad labour market conditions.

of Research and the Australian National University Humanities & Social Sciences Delegated Ethics Review Committee. Interested researchers may contact the Humanities & Social Sciences Delegated Ethics Review Committee via email at human.ethics.officer@anu.edu.au with data access requests.

**Funding:** The study was funded by the Australian Research Council Discovery Programme grant number DP 140102856. The funders had no role in study design, data collection and analysis, decision to publish, or preparation of the manuscript.

**Competing interests:** The authors have declared that no competing interests exist.

# Introduction

For more than four decades, the theory of job control and mechanisms for its effects on health–particularly coronary heart disease, mental health and musculo-skeletal diseases—have been under scrutiny. Over this same time period, the public health field has unsuccessfully sought to improve the population's diet and physical activity in order to restrict the growth in non-communicable disease (NCD) [1]. Despite considerable productivity loss estimates from NCD trends [2], the links between recent shifts in industrial relations policies and worker health remain under-explored with the focus being on firm-level management systems and job redesign. This oversight is perhaps the more notable given that flexible work arrangements are justified by governments as important planks of the productivity and the health policy mix [3].

A range of OECD countries has rationalised that time-flexible work policies give workers more control to help them manage work and family obligations, which in turn improves work-life balance, reduces stress and improves their health [4]. Yet, few studies have considered flexibility as a process whereby workers may shift their time away from health-related activities, especially when individual performance, and possibly job security, are tied to firm-level performance. Whilst there are multiple and complex barriers to disease prevention practices beyond the temporalities of working life [5, 6], there is clearly a need for research on the interrelationship between the new temporalities of working life and the temporal dimensions underpinning a range of health risks both directly and indirectly linked to job conditions. Direct links can take the form of the sedentary nature of jobs or physical hazards while indirect links include work-life conflict or work-induced exhaustion disrupting health practices including sleep, exercise and heathy diets [7].

In the Australian context, factors underlying poor health for employed adults include long hours in tandem with intense hours [8], with time at work identified as an impediment to time for healthy eating and physical activity [9, 10,11]: two health risk factors underlying coronary heart disease and other preventable chronic diseases [12]. The implications of recent work time transformations for job control, and the resulting impact on health promoting practices, form the primary aim of this study.

## Conceptualising control

In landmark work, Karasek [13] observed health and behavioural variations in jobs with different combinations of control over tasks and workload demands. He also observed that the possibility of using and developing skills was closely correlated to having a say over tasks and timing. Karasek argued that high workloads were a problem for mental health when they were coupled with an inability to control how much time workers had to complete their tasks, or when jobs were boring. High job demands alongside low control were also associated with cardio-vascular heart disease [14]. By contrast, heavy workloads, in a context of decision latitude, were viewed as desirable, and in terms of health, not harmful. Job control insights have provoked decades of research activity and management practices [15,16,17,18].

While Karasek considered the changing nature of jobs being ushered in by deregulated labour markets, his work pre-dated the full flowering of flexible employment and non-standard conditions [19] including the emphasis on employer-employee negotiations over time use patterns at work. For all types of workers, these changes have resulted in alterations to qualities of work time, including intensity (having to compress time to meet deadlines) and unpredictability to meet demand peaks. Such reorganisation has removed work-time boundaries leading to a loss of rhythm and regularity [20], especially for salaried employees where work is often elastic regarding time and place through the applications of mobile technologies [21].

Within a backdrop of these new employment regimes, and their associated challenges of "increasing time pressures, time speed-ups, and work-family time conflicts" [22](p159), Moen and colleagues [22] have elaborated a time strain model that operates independently of, and synergistically with, Karasek's job strain model. In their conceptualization, the quality of jobs are impacted by the experience of work-time demands inside and, crucially, outside of the job. Their time strain model involves measures of time demands (work hours, deadlines, perceived time pressure including too much work to do in the available time) and time control (sense of control over the foregoing factors, plus "time adequacy", including enough time for themselves and families). In their study of white collar workers, higher levels of work-time demands and lower levels of time control–especially adequate time to prevent home strain–were related to poorer health outcomes.

Moen and colleagues' study [22] was cross sectional and they noted the omission of blue collar workers as a study limitation. The study outlined below includes a mix of white and blue collar workers, and approaches the topic through qualitative research in order to capture more fully the lived experience of the multiple and intersecting dimensions of work-time, "time adequacy" and health promoting practices. The study specifically assesses the proposition that flexible work arrangements will enhance the likelihood of workers engaging in health practices. It provides a valuable "complementary source[s] of information" that can improve the interpretation of working conditions surveys [23].

## The Australian industrial relations system and possible links to health

Australia's industrial relations system has changed in fundamental ways over the past 40 years. There are now more tailored and diverse ways of working, including hours, which deliver greater control to workers and employers. First, the focus of bargaining has moved away from collective and industry conditions, towards individual and enterprise based negotiation [24](pp80–81). Second, trade union membership has declined to 15 percent of the workforce further individualising negotiations between workers and supervisors. Third, the demise of the full-time, male breadwinner employment model has been ongoing with the growth in female labour force participation rates and the expansion in non-standard jobs (part-time, temporary, on line, zero contracts) that often operate outside of national minimum employment standards [25]. Fourth, today's national minimum standards include the right to request flexible working arrangements for some employees (those with care giving responsibilities, older workers) including: reduced hours, changes to start and stop times, working from different locations, and carers leave [26].

There is a growing body of Australian work examining the costs and benefits of non-standard employment, casual employment and precarious employment [25, 27,28, 29]. Typically, the studies conclude that employee health and well-being suffer under each of these employment regimes. Receiving less attention is flexible employment, although several survey studies have considered this regime from a work-life conflict perspective [30, 31]. One study [29] on casual labour in call centres found higher levels of work-life conflict among those with greater work intensity, which was in turn associated with more fatigue and poorer psychological health. In terms of the focus of this study, a more recent study reported that flexible work may increase sedentary behaviour and have no appreciable impact on physical activity [32]. This finding contrasts with an older US study which found flexible work to be associated with supporting more frequent physical activity and more sleep [33]. A recent review of this literature concluded that it is not yet possible to discern consensus regarding the health and well-being effects of flexible work [6].

This finding may in part be due to the range of temporal aspects to job demand, which maybe mediated and attenuated by flexible employment policies. With flexible work

potentially encouraging long work hours [34] it is relevant to understand the pathways between flexible hours, long work hours and health practices. Similarly it appears relevant to examine the way in which flexible work can encourage work hours spill-over into family and personal life, amplifying stress [35]. Further, when job intensification and its associations with diminished health and well-being [10] is attributed to flexible work [29], it is pertinent to ask whether the flexible work regime bears some responsibility for this finding.

While a qualitative study cannot tease out the relative strength of association between employee control over their work-hour schedules and long hours, intense hours and work-life interference, it can provide deeper insight into how these job characteristics are interpreted as being interrelated by workers. Such an approach can also shed light on the efforts that employees make to reconcile the competing demands over their time, and in the context of this study in allocating time to some very basic health practices.

## Approach and methods

This study draws upon practice sociology which treats time as a core organising principle in people's lives and in social institutions [36, 37]. Practices such as working, eating and daily exercise are thought to emerge out of performing tasks necessary for survival, for pleasure and other motivations. They require individual and social resources, especially time, as well as basic levels of competence and material assets [38]. Time is understood as having three important elements: routines, rhythms and duration. Routine describes the repetitive undertaking of tasks, involving the unreflexive sequencing and coordination of elements to the task [38], resulting in efficiency of effort [37]. Given that most practice routines involve coordination with, and accountability to others, the notion of rhythm becomes salient. The daily rhythm is an alignment of numerous social practices with one another in space and time, based on, and for the purpose of, social interactions [36]. Daily life rhythms can be upset when "The trajectory of any one practice . . . affect[s] the trajectory of others" [39](p41). Finally, duration refers to the amount of time practices take, typically measured in terms of hours and minutes. Duration is fundamental to understanding the reach of work demands into other practices and vice versa; and the way short intrusions from work may not disrupt routines or alter rhythms in the same way lengthy ones do. A small body of work is identifying the same temporal elements—routines, rhythms and minimum duration–as underpinning health promoting food and physical activity practices [40, 41].

Practice theory's utility extends to revealing how single practices are contingent upon a host of other practices, due to their temporal interconnectedness. The conceptual starting point to this study is the notion that the labour market affects workers' health through competition over the time they can allocate to a suite of everyday practices, each of which is time dependent [42]. The study also examines broad occupational status differences in practices due to flexible forms of work. Twenty five years ago, a Belgian study found that flexible work policies required manual workers, unlike professional workers, to adapt their value commitments and ways of living (e.g. deferral of gratification) to the temporal demands of their jobs in order to gain employment [43,44]. In line with Karasek's theory, low status occupations were subjected to greater employer control over working conditions and had flexibility forced upon them, unlike higher status occupations who could better control their conditions (rosters, predictable days, contracts, task autonomy). The study concluded that the adoption of a culturally flexible disposition was as a feature of disadvantage while the capacity to maintain valued approaches to everyday life was a marker of social advantage [43, 44].

For this study, we approached two firms from the construction and insurance industries with different enterprise agreements to recruit participants for semi-structured interviews.

This approach maximised the diversity of employees (women and men, representation across occupational categories–manual work, service/care work, management) and variety of working time arrangements (long hours, part-time and night time shifts). Participants agreed to join the study in response to a letter the study team sent to worksites (S1 File) inviting participants to an interview during working hours with the managers' agreement (S2 File). We aimed to recruit a non-representative sample of around about 12 participants at each worksite based on research showing that new relevant information is rarely gained through additional interviews [45]. Written consent was obtained prior to interview and confidentiality assured, given managers knew who was participating; with participants able to exercise their rights of withdrawal (S3 File). A total of 28 workers was recruited, 13 at the construction worksite and 15 at the insurance worksite. The study was approved by the Australian National University Science and Medical Ethics Research Committee (Approval 2014/285).

Prior to interview, participants were provided with a package which included study information, a consent form and two time diaries, one for a week day (S4 File) and one for a Sunday (S5 File), accompanied by a request to record time spent on a range of activities—work, eating, physical activity, sleep and leisure–immediately prior to the interview. The diaries were adapted from the Longitudinal Study of Australian Children (LSAC) "lite" time use diary [46]. The diary, as well as purpose-designed "time tools", were used as prompts during the interview, with participants asked to rate dimensions of work time control (S6 File) and flexibility (contained within S7 File). The interview was loosely structured around a list of open-ended questions (S7 File) and followed the asking of background information, including questions of job time strain (S8 File). The data enabled a comparison of the two workforces across both worksites. In recognition for their efforts, but without their prior knowledge, participants received a $50 voucher to a grocery/department store at the end of the interview.

Lasting between 45 and 90 minutes, the interviews were conducted from February to August 2015, and were recorded and commercially transcribed. Interview transcripts were coded and sorted thematically by three of the interviewers using Atlas TI software. Coding categories were developed based on 1) relevant theory and research findings on working time, labour market, health practices (food, exercise), and other everyday practices like child care, and, 2) iterations of immersion in the transcripts. All team members contributed to the thematic interpretation of the interviews alongside the workforce demographic data and administered time tools. To validate the team's interpretations, participating organisations received a summary of findings for their worksite and were provided with an oral presentation at a specially convened staff meeting. At the presentation, those interviewed and other interested staff could seek clarification or modification of interpretations. There was one substantial instance of the latter at the construction worksite. At the end of this process, a policy brief containing major study findings and policy recommendations was posted on the study site.

In what follows, the names of the two work sites and all study participants have been given pseudonyms.

## Findings

The blue collar site, Altor Construction, was a multinational corporation with a quarry located near a small inland town in Victoria. Approximately 60 workers were involved in earth moving, processing and distribution of construction materials. Most lived locally and had short commute times, though several commuted up to 1.5 hours. Work was generally semi-sedentary and involved heavy machine operation. Altor manual worker rosters were negotiated between the union and employer providing workers with pay for up to 11 hour days (8 hours on basic pay, plus penalty rates for additional hours). Most worked 52 to 55 hours a week. The

only two women on site were employed in administrative positions, while two men were employed in management and two as self-employed contractors.

The white collar site, Drake, was an insurance provision company located in a large regional town in Victoria. The company, with 200 employees, had an on-site call centre and most jobs entailed sitting in front of computers or attending meetings, with call centre work being more sedentary. Commutes to work were generally of short duration although one travelled over two hours daily. Work time flexibility was written into company policies, was widespread outside of the call centre, and occurred at the discretion of supervisors. For these white collar workers, not being on industrial awards which specified maximum hours was an enticement to very long working days that were often equivalent in length to those of the construction workers. The professional employees also worked under a Key Performance regime where individual performance was assessed on annual negotiated goals and targets.

Table 1 sets out the key demographic, employment conditions and self-rated health characteristics of the interviewees in both sites.

Comparing the demographic features, as well as their workplace decision-latitude cultures, self-rated health scores and job satisfaction scores shows Drake insurance workers to be younger, to have considerably higher educational status, and higher input into decisions and actions; each of which potentially explains their higher self-rated health (66% good/very good) against Altor workers (38% good/very good). As might be predicted by Karasek, Altor Construction workers had less input to organisational decisions but they reported slightly higher "adequate recognition from supervisors". A majority of households in both sites had children living with them, although Drake household children were younger and hence more demanding of care. Two Drake female employees with young children worked part-time.

**Table 1. Demographic, work and health characteristics of interviewees by site (n = 28).**

|  | Altor Construction (n = 13) | Drake insurance (n = 15) |
|---|---|---|
| Age: mean (range) | 48 (33–65) | 37 (27–53) |
| Female n (%) | 2 (15) | 9 (60) |
| Born in Australia n (%) | 10 (77) | 12 (80) |
| Partnered n (%) | 10 (77) | 12 (80) |
| Resident children n (%) | 9 (69) | 11 (73) |
| *Highest education* n (%) |  |  |
| $\leq$ Yr 12 | 3 (23) | 2 (13) |
| TAFE certificate/diploma | 8 (61) | 2 (13) |
| Tertiary | 0 | 11 (73) |
| Job types | Admin/management/machine work | Management/ sales/call centre |
| Years in job: mean (range) | 3 (0–12) | 3 (0–15) |
| Part-time work n (%) | 0 (0) | 2 (13%) |
| Adequate recognition from supervisor | 12 (92) | 13 (86) |
| Input into decision and actions | 8 (61) | 13 (86) |
| Job satisfaction: mean (range: 1 extreme dissatisfied -7 extreme satisfied) | 5.4 (3–7) | 5.6 (4–7) |
| *Self-rated health* n (%) |  |  |
| Poor/Fair | 3 (23) | 2 (13) |
| Good | 5 (38) | 3 (20) |
| Very Good/Excellent | 5 (38) | 10 (66) |

Comparing access to flexible work provisions (see Appendix 7 for the protocol used) indicates Drake workers to be more likely to access most forms of flexibility (start and stop times, working location, reduction in hours, vary work days), although very high proportions of workers in both sites could leave work to manage unexpected emergencies or family needs. Few blue collar workers reported the option of varying start and finish times over the long term, and changing workdays was highly unlikely as was reducing hours or reducing responsibilities. These limited conditions applied also to the two Drake call centre staff. Unlike Drake white collar workers who had access to location flexibilities, working off-site was impossible for their call centre staff and for Altor machinery operators and managers.

Three flexibility indicators used–can take extended leave, reduce responsibilities and work-off site—can be considered "time adequacy" indicators, in the Moen model, and heavily favoured Drake professionals. The interview data, however, provide a more nuanced picture of who benefitted from these conditions and could explain why, despite their greater access to FWAs, Drake worker mean average "job satisfaction" (see Table 1) was only modestly higher than Altor's.

In order to examine the time strain and health interplay we combined data across both sites in terms of workers that have, and do not have, "control over start and stop times"–the most basic indicator of flexible working conditions under Australia's Fair Work Act. See Table 2.

While the high flexibility group, which included four Altor workers, had greater job control—in terms of ability to reduce responsibilities and input to organisational decisions–they scored lower on all three job demand factors–workload is reasonable, can complete workload in regular hours, and works to very tight deadlines. However, they did not feel rushed or pressed for time to the same extent as the low flexibility group (in particular the machinery operators), explained by being able to extend working hours and taking work home. Consonant with Karasek's theory, the higher flexibility group had higher demands, greater control and decision authority, and higher self-rated health. Their narratives nevertheless revealed high levels of stress, counter to Karasek's theory, and the interviews highlighted the tensions that having flexible work conditions brings: workers with higher temporal control reduced some temporal demands (being rushed or pressed for time) by exacerbating others (longer worker hours), and fostering feelings of time inadequacy especially regarding personal time and home life strain [47].

## Qualitative research findings

Following the distinction between low and high flexibility employee groups as reported in Table 2, we now use the interview material to tease out the implications of having differing

**Table 2. Worker perceptions of time strain by degree of flexibility (n = 27[a]).**

| | Low flexibility Group A (n = 11) | High flexibility Group B (n = 16) |
|---|---|---|
| Rushed or pressed for time (Mean Range: 1 always—5 never) | 2.9 (1–5) | 2.2 (1–4) |
| | Number (% yes) | Number (% yes) |
| Workload is reasonable | 10 (91) | 14 (87) |
| Can complete workload in regular hours | 8 (73) | 10 (62) |
| Works to very tight deadlines | 6 (55) | 15.5[b] (97) |

[a]We are missing time strain data for 1 worker.
[b]Includes answered 'sometimes'.

degrees of temporal control over various aspects of work demand for the two health practices which are the focus of this study.

**Group A: Low worker flexibility.** Group A, Altor Construction manual labour and administration employees and three Drake workers, had low flexibility because they had no capacity to negotiate their rosters. Indeed employer control over work rosters is a feature of life for low occupational status workers [29]. However, clocking on and off at the same time in each day delivered them predictable daily, weekly and monthly schedules enabling other practice routines and rhythms.

Typical of this arrangement was Barry, a 42 year old Altor Construction employee with a trade certificate who, having left school at 15, worked 11 hour day shifts. Barry was on the preferred shift, arriving home at 5pm in time to prepare dinner. Barry said he cooked for two reasons: his wife, a school teacher arrives home an hour later than him, and is "not the greatest of cooks". They eat usually at 6.30 in front of television, and go to bed at 8.30 or 9pm given that his shift begins at 6am. His TV watching habit is justified in the following terms: "by the time we get home and all that we're both knackered".

While they had predictable and routine work schedules, their long shifts were the key factor affecting the Altor Construction workers' approach to health. Machine operator Hal echoed the other men's common experience of exhaustion: "I enjoy walking and all that, but I don't do it that often because I'm just knackered all the time". Addressing Moen's "time adequacy" for personal life, Luke, a 33 year old machine maintenance contractor with Altor, rostered on from 4.30am to 3pm, spoke openly about how his working life–both the long hours and intensity of the demands (he is responsible for fixing broken machines so that night and day shifts can remain operational)–had contributed to anxiety. "Yeah I get anxiety, you do get anxiety, but I didn't realise I got it until— . . . I was sitting at my mum's place, and it just come on out of nowhere, I just had to leave, I couldn't be there, I felt the walls were closing in". He emphasised his weekly fatigue–"By Friday I'm not real good to communicate with"–and how that led to loneliness. A young single man, Luke would have rated at the extreme end of time strain and by his own admission his eating practices were 'shit' and he relied on the high energy soft drink Red Bull to get through his shifts.

The majority of men, including Luke, described being highly physically active on weekends, often in the form of a hobby like shooting animals, car racing or mountain trail bike riding. However, these activities were irregular and depended on weather, exhaustion and family commitments. The less physically onerous family meal seemed to follow distinct patterns because workers knew when they would be leaving work and they did not bring work home. Barry was not alone in enjoying to cook, with a number also enjoying the garden but not other home duties. Most had teenage or adult children allowing their wives to work full or part-time. Single father Kevin, 46 and a machine operator lived next door to his parents and noted. . . "if it wasn't for my mum" everything would fall over to do with "the kid".

Whereas a rigid shift helped a few Altor workers engage in regular physical activity and all had set meal times, a rigid roster is not sufficient for others. Debbie, a Drake insurance call centre worker, with hours set by the employer, said that her three day a week call centre job interferes with her attempts to keep physically active: the 8.5 hour daily roster stipulates she sits for two hours before a 10 minute break. She described the work process as totally inflexible and "boring" unlike her previous more interesting, but stressful, professional position. Drake management would not let Debbie job share in her original position so she could reduce her days to care for two small children. Her husband had highly unpredictable hours and his long days, and lack of routine finishing time, coupled with her rotating shifts, meant Debbie found it hard to maintain any sort of personal routine. The one exception was that she always had Mondays off, when she fitted in a "big walk" plus housework; she took meal planning very

seriously so that the family did not eat take-away food. Her late rotating shift which included a Thursday and Friday evening meant that she could not attend her son's footy training and her children's swimming lessons. She reflected, "I don't like missing out on my kids" things".

The two women from the Altor Construction office had predictable days too but of shorter duration than the men's, and they had access to rostered days off when accumulating overtime. They maintained a more consistent routine regarding their food and physical activity than most participants. They did not have children at home and had reached an age where they said that they prioritised their health unlike the men of a similar age. Vanessa, 54, exclusively ate prepared weight loss foods and had recently lost 20 kilograms, in a bid to get fit so she could travel in retirement. She "loved" her job. Sophia at 65 was equally enthusiastic about her job. Her husband cooked, they saw a lot of their adult children and eight grandchildren on weekends; and during the week Sophie had an active social life, made possible because she finished at 4pm each day. Her physical activity consisted of walking. Both praised their supervisor's attitude towards granting time for personal matters, especially for Vanessa who attended to her elderly parents' appointments. They were possibly the most satisfied with their lives of all 28 interviewees.

**Group B: High worker flexibility.**   Group B, largely Drake Insurance professionals and managers and two Altor plant supervisors and two self-employed contractors, had higher control over start and stop times. Drake workers could also elect where to work (home or office). For the majority the flexible arrangements delivered unpredictable days with daily, weekly and monthly hours being driven by firm-driven output requirements: the financial reporting cycle; roll-out of new programs and staff hiring; and human resources training. While they had greater say in organisational decisions than Group A, they often worked to very tight deadlines.

It was common for the professional and managerial workers to start at the same individually determined time each day but to vary their finish time according to the highest priority task that day, which might be completed at home, a finding that challenges the extant notion of job control. For these higher status and seemingly autonomous employees there was a blurring of boundaries between worker and employer control. Workers could control the sequencing and content of the work but the workload and priorities were often unpredictable and outside their sphere of control. In effect, theirs was a compromised control. Within Group B, there was a small sub-group who stuck more rigidly to a specific finish time (termed the resolute control group) and we examine their experience separately to the compromised control sub-group. Moen's time adequacy concept would suggest that Group B workers have greater success in establishing health practices than Group A workers because of greater control over personal time, while practice theory would suggest that those workers without sufficient work routines would struggle to maintain non-work practice routines.

**Group B: Compromised control sub-group.**   Warren, 30, finance manager at Drake, described having 10 to 12 hour workdays, especially at peak periods when the accounts were due. He could determine when he arrived at work and when he left but the amount of work he had to do remained resolutely high. He explained his daily eating practices in the following way:

> I've had two coffees, does that count [for breakfast]? I always eat lunch. . . . Often it's here [at the desk], which I think's probably far too common. I've been trying to make an effort . . . just to get myself away from the desk. . . . I generally have a pretty healthy lunch. . .We're doing Lite'n'Easy at the moment [for dinner], which is a convenient / health decision . . . it's not all week, but it's certainly the weekdays, . . . on that point, with my son growing up, I really want to have a time where we sit down at the table as a family, at dinner. Because

that's what I did growing up and so . . . that's something that I want to move towards, sooner than later.

Warren's physical activity was more routine than his eating: he parked at his mother-in-law's house so he is forced to walk into work (15–20 minutes). He said work hours undermined his desire to do more activity. Warren's health practices were remarkably similar to those of John and Martin, each with upper senior management positions at Drake and young children. All three were inserting physical activity into the working day by getting up early, prioritising active transport, and doing as much physically as they could fit in on weekends.

Drake Insurance employees were less sanguine about missing out on physical activity than Altor Construction workers. As Martin put it when describing his heroic efforts to fit in exercise around work: "[Regular exercise]: It's your mental health. It's your physical health. It's everything". Here one senses that workers view physical activity as offering protection against job strain, and are engaged in effort-recovery practices.

The setting of start and stop times by professional and managerial workers does not protect them against exhaustion, however, especially when they take work home because of the extension to the working day. In this situation, it was irrelevant who set the timing and duration of the working day. Only one half of Drake insurance workers could complete work in regular hours and they expressed a desire for limits over work intensity and work volume. Rosalie voiced a sentiment shared by several of her colleagues: "I constantly feel too busy. My main thing is that I don't feel like I do anything particularly well because I'm too busy to actually focus on something and do a really good job of it. I just have to get it done".

Compromised work-time control was evident in the accounts of the Altor Construction managers and contractors; none of whom were bound by the Altor enterprise agreement. Like Drake professional employees, they worked a mix of paid and unpaid overtime. Tim was a self-employed contractor leasing equipment to the Altor Construction quarry. He set his own hours, doing a minimum of 12 hour days with unpredictable finishing times. His eating and general state of exhaustion were not what he wanted. "It's a lot of office work these days and driving, a lot of driving. On the phone heaps. It sort of starts at around 5:30 to 6:00 o'clock in the morning. It used to be earlier but I find I'm working a lot later of a night now, so normal knock off is anywhere after six. I'm trying to keep it to a 12 hour day with family and that".

As to his food routines and their content: Breakfast: "on the run, Maccas [McDonald's], just a raisin toast and coffee". Lunch: "Head up to the food van out the front . . . I'd probably have a salad roll with chicken. Something like that. I eat quite healthy". Dinner last night: "I was that buggered. I was trying to get them organised and they [children] were mucking around. I just went to bed [without dinner]".

As Warren and Tim demonstrate, when flexibility provisions foster unpredictability in work time schedules and output as well as long hours, they can undermine healthy eating and physical activity practices that involve a predictable rhythmic quality. Moreover, Warren reflected that he did not feel his current work-family juggling act was maintainable. Without prompting he revealed "It's just not probably a sustainable level of effort . . . at the minute. If I was to really ask the hard question of, do I think it's worth it? It's probably not, to keep going. But, at the same time, I'm young so I am happy to have a good crack. But it's got to be the right balance".

Another male manager at Drake with two young children reflected on whether he had enough "free time". Martin said he was satisfied for himself–he was a dedicated marathon runner outside of work–but he wanted to be present when the children got home from school and he wanted more time with his wife. He reflected that ideally: "you wouldn't be away from home for eight, ten hours a day would you?" The experiences of Warren and Martin resonate

with evidence of a relationship between high work demands and emotional exhaustion [47]. For Warren and his managerial colleagues, the key performance indicator movement which underpins their employment prospects (promotion, tenure) exerts pressure for potentially exploitative work time regimes that flexible work regulations compound, especially when the ceiling on daily/weekly working hours is removed.

**Group B: Resolute control sub-group.**   A minority (6 out of 16 workers), working under a flexible work arrangement across the two firms, appeared to adopt a rigid daily schedule (approximately the same start and stop time) with only the occasional incursion into home and leisure time. However, they had higher on average "rushing" scores than their peers. What unites this group is unclear, although the bounded work time schedule was justified in terms of care of children in four instances—one single mother, one mother with a child with special needs who had cut back to four days, one father of four children and one mother who also had a health condition that she was attending to. Children can be seen as a competing devotion that provides a strong counter to work performance expectations. The two child-free workers were older and financially secure.

Five of the six resolute sub-group exhibited healthier practice routines, and it could be argued that they were adopting a culturally rigid disposition as proposed in the Belgian study for high occupational status workers. However, not all belonged to this occupational status category. David, a 42 year old boiler maker with a Trade certificate, described his employment with Altor Construction as following predictable hours. David's was a remarkable tale of family member synchronicity, with the side-benefit of a good deal of physical activity. With four children he had adjusted his hours to start at 6am in order to finish at 4.30/5pm to accommodate his sports-loving children's after school activities.

"And then Tuesday night I got two kids that have football training and daughter at netball training and on Wednesday we got another boy that trains in B. And then my little boy has footy training Wednesday night. . ., so the wife takes him there, and Thursday night there's two lots of footy training and one at netball training and the youngest one goes swimming . . . Yeah, so it's all time management. I'll go swimming from five to six in M. when I'm there".

The presence of children could not explain all of the resolute control behaviour. Curtailing long hours to fewer than 45 hours, in concert with predictable hours, appears integral to the stories of Anna and Gus, Drake professionals who were child-free. Each adopted a repetitive and predictable set of health-life-work practices. Anna 47 was the most dedicated of our 28 interviewees in adopting health promoting practices, with highly routinized eating and physical activity schedules synchronised with her rigid work schedule. Her husband cooked dinner because he got home first. She arrived and left work within 15 to 30 minutes at the same time each day, describing her days thus:

> I have breakfast at work. . . .I turn my computer on,. . . I bring my breakfast that I'd prepared on Sunday. So, I generally make a vegetable frittata or something on a Sunday and have that all portioned out. . . So, in my calendar, I have 12:00 to 1:00 booked for lunch, and I do that because I am someone who can just work and work and work and forget to eat, or forget to have a break. . . I generally leave here at about five, 5.30 and I go straight to the gym on the way home. . . . So, I get home about seven, 7.15. We have dinner. I ring my mum, because she's not been well so I ring her every night. We then take the dog for a mini walk, maybe 15, 20 minutes. Come back, relax, shower, go to bed and it starts again.

The resolute control worker who was not successful in pursuing her desired health practices despite a self-determined rigid work time routine was single mother Sue, 37. A project manager at Drake Insurance, Sue was typically in the office between 9 to 5pm and rated herself as

almost always rushing for time. While she set these predictable hours, she argued that her present full-time job was the major deterrent to getting sufficient exercise and having a social life. Even though she described herself as driven by routine due to her nursing background, she compared her current situation unfavourably with her previous three day a week job. "I was exercising and had things like, I suppose me-time, so I was able to go to the gym while kids were at school. . . . [now] Even catching up with some friendships, I found that really hard to maintain, because of the hours, the nine to five, and the school mum thing".

She wished to do more physical activity "Just the exercising part really. And [work] doesn't really interfere, that's my choice isn't it. Yeah, because I could get up earlier". However, Sue was very clear that establishing any health routine depends on multiple factors including: the job hours, children's sporting commitments and social events. Like many of the young professionals, she was adamant that she could not hold her life together without the support of her parents.

**Common finding: The limited use of flexible work provisions linked to job insecurity.** In both firms, negotiated flexi-time was used by all participants for medical appointments, urgent errands and emergencies such as caring for sick children or elderly parents. If the Altor Construction workers did use flexi-time for an appointment or emergency they did not feel that they had to work harder the next day. By comparison, the service sector workers felt they had to catch-up on yesterday's work before tackling new tasks, indicating how such conditions offered only partial control over work time. Flexi-time was generally understood not as a right but as a partnership arrangement; as one senior Drake insurance manager put it: "we run lean here, lean with high workloads. . .it's a give and take system. I don't mind giving, because then I feel comfortable taking it when I need to".

The discretionary nature of the flexibility system was apparent in both firms, but especially so at Drake which operated on the basis of a tight fit between organizational and individual performance, as is emblematic of the KPI ethos. Drake manager, Rodney, responsible for granting flexibility requests from his team revealed the compromised control nature of both the flexibility and key performance systems: "Now I do say I believe in workplace flexibility because I always win. You will always do more for me than I give back because that's the type of people you are. You're committed to the organization as well. . .this place runs on unpaid overtime. But most organisations do". Unsurprisingly, Rodney noted the high importance of work in his own life."Should I spend more time with [my kids]? Yeah Sure. Should I do more exercise? Definitely. Everything gives. Except work. Work comes first".

Altor Construction workers were more explicit in attributing their limited use of flexibility provisions to job insecurity as opposed to company loyalty. They were aware that their jobs were predicated on macro-economic forces–principally, the global price for minerals—and that Altor Construction was closing down their low output quarries. Stephen, commented: "I wouldn't say the job is secure. It's just the way the [quarry materials] environment is these days."

Fear of being replaced encourages workers to comply with demands to work longer hours and not to question management decisions. For the Altor Construction machine operators, feelings of insecurity were also associated with fear of losing their overtime pay. Some said they did not have the financial reserves to turn down extra hours. Archie described himself as "overtime hungry" because he was "financially, just trying to get on top of things".

A resignation about poor health was palpable. All Altor workers could articulate a good diet and physical exercise as important to their health, but preventive health practices ranked lower than job satisfaction and a decent pay packet that accompanied their 55 hour weeks. In answer to keeping fit and relaxing, Altor machine operator Barry replied: "I'd like to be doing more on that side but it's just—Work's number one priority and . . . it's on the backburner the

exercise and all that sort of stuff . . . Unfortunately it is a high priority but there are more things [paying the bills] higher than that". Luke, 33 years, with the "shit" diet, Red Bull habit, and anxiety attacks recognised that he was not healthy: "Bloody hell, I'll be lucky to make 60". Employed on rotating night shifts, Archie 38 said he didn't do anything to stay fit or healthy. "Not that I don't care. . .". He tells his wife when she worries about his tiredness, "I'll sleep when I die".

In contrast, Drake workers noted their firm was expanding and they were less concerned about broader economic conditions. They were younger and more inclined to switch jobs if the pressure of the current job led to undue strains on the family. Some with permanent jobs felt very secure, but their career progression depended on working additional hours to the standard working week. This hidden requirement to deliver on time-extensive and unpredictable tasks undermines the reality of worker control. The greater temporal autonomy–in terms of flexibility provisions—of permanently employed Drake insurance workers was clearly not sufficient to guarantee time for health.

Indeed, the experience of job insecurity, coupled with a lack of predictability around schedules among our relatively privileged workers appears to be approaching that of Australian casual workers where the more extreme lack of certainty around schedules and job security was accompanied by "debilitating consequences of workers' autonomy, self-efficacy and control over their lives" [27](p766).

**Common finding: Need for support from partners and extended family.** The maintenance of health practice routines for long hour workers in both worksites requires a household partnership. For women this typically means putting their own careers on hold: they either remain unemployed until the children attend school; hugely limit their hours to accommodate their partner's work (the wife of Warren from the compromised control group who works Saturday morning only); take a lesser job with reduced hours (Emma below); or rely on parents or extended family to undertake considerable care duties (as with Debbie below). Thus, the capacity to exercise job control can be viewed as a gendered process that relies on the capacities of linked others.

Emma used to work for Drake Insurance five days a week as a process analyst, a job she found satisfying but stressful because it had lots of components and deadlines. When her husband got a new job with a long commute, she approached her employer about cutting back to three days. Drake responded that the only three day job available was in the call centre, which meant a rotating shift of 9–5.30 some weeks and 11.30-8pm other weeks. She rationalised the situation as having costs and benefits.

> It mainly suits me. I've got two young children, I've got a one year old and a ten year old. So sometimes it's hard with the late shifts with school pick-ups and things like that but then again I don't mind doing the late [shifts] because it means that I get some time in the mornings where I've got time for myself. So that's nice. . . . On an 11:30 till 8 shift my partner sometimes cooks. It's rare but he sometimes cooks and I either catch up with a friend for coffee in the morning or I'll do some housework in the morning, it depends how behind I am.

Emma appreciates having less stress but said that her new job was boring. In responding to the observation that looking after the home is a big commitment, she reflected: "Yeah, that's true. Yeah, my partner's a bit, I'd like to say old fashioned but really sexist and as far as he's concerned he earns the money and I do the house stuff, even though I work too but he's a bit stuck in the past that way".

Her husband's long days and unpredictable finishing times, coupled with her rotating shifts, means Emma finds it hard to maintain any sort of personal routine. The one exception is that she always has Mondays off, when she fits in a long walk along with doing weekly meal planning and housework. Her late rotating shift on a Thursday and Friday evening is viewed as not ideal because she misses out on her children's swimming lessons and football training.

For single parents, close proximity with their parents who can undertake child care operates as another form of partnership. Debbie, 37, from Drake Insurance, described her dependency on her parents and resulting guilt in the following way:

> [My parents] step in and help out. So they're great with my kids, and so mum takes them to school every morning and picks them up every night. So yeah, so it's huge, yeah. It saves me a huge amount of money too, with this care. . . . Last week my daughter was sick and like I said . . . my mum will drop everything to help out. But I kind of felt that because she makes plans from the nine until three, that she put[s] those on hold because I chose to come to work. . . .I'll always choose work, because it pays the bills, basically.

Single father, Barry, a machine operator for Altor Construction lived next door to his parents and noted 'if it wasn't for my mum' everything would fall over to do with 'the kid'.

Employer designed work schedules generally operate as though workers are individuals. There is limited acknowledgement that workers are embedded in social relationships. These relationships both support, and place demands on, individual workers in ways that are unacknowledged or unrecognised by workplaces at a structural level, even though "bosses" or managers will make concessions for workers on an ad-hoc basis, such as when a child is sick. The societal and workplace expectation is that workers will organise their lives so that these external relationships will not impinge on work time. Partners, usually women, often make these accommodations possible.

## Discussion

Following practice sociology, this small qualitative study was designed to explore how the temporalities of two major health practices are being shaped by new flexible forms of employment. As a result of the interviews, the key performance indicator (KPI) movement–with its control over the working lives of professional workers, managers and self-employed contractors–became too significant too ignore. Through the individualisation of effort which is integral to this movement, workers' goals are tied to organizational goals thus shifting responsibility for firm level outputs away from a collective effort toward individuals [48, 49]. The KPI movement deploys measures of worker productivity as a key mechanism of employer enforcement and as a basis for the rewards system (pay, promotion). Under this scenario, performance output expectations replace time bound days and weeks, especially for professional and managerial positions. Such a work-time regime is in part responsible for what has been described as the "precarization" of managers [50].

In an early piece, Karasek [51] plotted the position of various occupations within the four quadrants of low control-low demand, low control-high demand, high control-low demand, high control-high demand. Fitters–the equivalent of our machine operators–sat in the low control- demand quarter and professionals in the high control- demand quarter. A major finding of this study is to question the neatness-of-fit between particular occupations and job control-demand. What our material indicates is that there has been a quiet shift underway due to the increasing impact of temporal demands over the intervening decades. Today's fitters, our construction workers, continue with low control but are subject to days of longer duration

(from 8 to 11/12 hour days) and hence they are now in the low control-high demand group and the professionals have shifted in this direction too due to the insertion of additional components of time strain: namely, not being able to complete workload in regular hours and working to very tight deadlines. This finding of the increasing ubiquity of low employee control over temporal demands that have consequences for health practices and well-being, like long hours and work intensity, suggests that the employer is firmly in control even when employees have input into organisational decisions. Second, the control over time that is associated with flexible work arrangements is circumscribed by other employment relations policies, namely an absence of labour market wide maximum hour working weeks and individualised performance regimes.

The fear of job loss was present among those with higher job control as well as lower control and had the impact of also encouraging long hours as well as time pressured hours. This finding accords with a pervasive sense of job insecurity in Australia; a trend that persists despite little change in job turnover [52]. For the professional/managerial workers, perceptions of job security mediated requests for greater flexibility. Irrespective of the nature of job tenure or contracts, workers have been "rendered subjectively precarious by the increasing power that senior managers wielded over them" [53](p451); and our study supports the conjecture that class differences in job security have diminished [54]. Instead of adopting a personal future orientation with health at the centre, with the exception of the small resolute control group most workers appear to be adopting short-term horizons which is manifest in: not asking for time off, not limiting work hours, and a fatalism among the manual labourers at least in regard to health promoting practices. The absence of a future orientation among these workers can be explained by their lower socio-economic status [55] and the fact that they were employed in an industry subject to downsizing [54].

In terms of the interplay between work time demands and health practice time demands, there was some protective effect of having a fixed schedule as opposed to a flexible one. It was less relevant who set the schedule: the employer or employee.

For Group A–primarily machine operators, low level administrative workers and call/service centre workers—with limited formal entitlements to workplace flexibility, there was less control over organisational decisions as would be predicted by the Karasek thesis. Typically, this group had predictable shifts, long hour days and working weeks (greater than 48 hours) leading to high levels of exhaustion which undercut what was conceivable during non-work time, especially physical activity. On the health promoting side, their predictable time slots combined with roster-determined meal breaks appeared to foster more routine meal patterns at home and at work. This finding accords with a large European survey which showed "having more fixed working hours makes possible a more stable regimen of basic physiological functions (e.g. sleep and meals) and a better planning of everyday life in terms of social integration with family members and family activities" [56](p1135). For Group A workers the home-work boundary was maintained although this did not stop many from expressing a desire for more time with family and for leisure. Those on shorter working weeks, had either chosen the particular position to dovetail with another job or because of limited job prospects, or in the case of women with young children and no supportive partner this was a "forced" choice.

For Group B–professionals, managers and self-employed contractors—limited control over the volume of work and high performance expectations led to long days organised for the majority around unpredictable time slots. These seemingly more privileged workers worked almost identical hours to the 11 hour quarry shift workers, but the latter were paid for all of their hours. For a majority of Group B, flexible work provisions were eclipsed by their performance contracts, a regime of individualised work agreements governed by performance

measures set by the employer. Under these employment arrangements, increased autonomy over working time has not been accompanied by a reduction of individual working hours as some have equated with flexible work [50].

Contrary to the Elchardus thesis, the professional workers were less rigid than might be expected regarding their health practices especially routine meal patterns; although a minority adopted relatively rigid work days and meal and physical activity routines. This resolute control subgroup largely confirmed the proposition that higher flexibility (as in individual autonomy) and lower variability (degree of fixed work timetable) are pre-requisites for better health and well-being [56](p1135).

Higher socio-economic status may no longer be protective of preventative health routines because the mandated jpb control is narrowly conceived around start and stop times, shift rostering and flexi-place. Flexible work is in effect a fluid system of limited control, endlessly negotiated between the employer and worker, with pressures from managers eroding work time autonomy [57]. It has meant greater fluidity is creeping into the boundaries around work and home, and a normalisation of expectations around this. As the person responsible for health and well-being at Drake Insurance stated "it's not so much work/life balance anymore, it's work/life integration, which can be good and can be bad".

Thus even though this higher flexibility group was the contemporary equivalent of Karasek's active group—challenging jobs with high workloads but good health prospects–they struggled to find time for their health under the new flexibilised and KPI conditions. In this respect this study sounds a warning: flexible work regimes which foster unpredictable schedules coupled with the KPI movement, which fosters job insecurity, can encourage not only an unrelenting annualisation of time but a reorientation in personally valued social and personal commitments, including protecting health.

Finally, in line with practice theory, the presence of children is a mediating factor; and appeared to affect men and women differently. Protecting time for care could be the main reason women accepted less favoured jobs with predictable rosters or it could provide the motivation to exercise work time control, as negotiated with the employer (and see [58]). In general, women's care of children was described as robbing them of time for health whereas men described adopting a physical activity routine as a way of spending time with their children. Gendered practice hierarchies have evolved based on a process of temporal displacements, which are in turn based on what is perceived as essential in the here-and-now and what can be deferred, or as others have observed "Routines for everyday decisions are a compromise between what is desirable and what is practical in given settings" [59](p127). Paid employment is taking precedence over other life activities, and in the context of weakening social safety nets, investigating who controls the terms, costs and benefits of different employment regimes remains an important question.

## Strengths and limitations

Our use of practice sociology turned the spotlight on detailed daily work and domestic routines shaped by work-based temporal constraints. The use of time diaries that included a work day and a weekend day allowed us to compare participants' measured work and leisure activities with their subjective experiences recorded during interviews. During interviews, participants reflected on how typical their diary records were and discussed unusual events. The strength of the study rests predominantly on employee experiences and explanations, reflecting the specific workplaces in which they work and their individual social circumstances rather than measures of time-use.

Due to the limited sample size and our qualitative approach it is not possible to generalise the findings, although the findings are broadly supportive of population based surveys that

show a decline in workers' temporal control. It adds to the small body of research that reveals the importance of work time for health practices outside work by generating three hypotheses for further consideration. First, decision latitude within a context of flexible work agreements alongside key performance indicators can foster time for health practices only when predictable time slots are an option and long hours can be contained. Two, the capacity for workers to set their work hours needs to be matched with regulations giving workers control to set limits around work volumes, deadlines and intensity. Third, a sense of job security can foster time for health practices. The presence of children and gendered differences in the each of these dynamics require consideration [58], especially given the role played by work hours in gender inequality in employment.

## Conclusion

This paper argues that current transformations in industrial relations—where work time has become unpredictable and unbounded—raises new and pressing questions about extant theories of job control and its impact on health. Our starting proposition was that workers who accessed flexible work conditions, including control over start and finish times, could incorporate health promoting activities within the work-life balance that was supposed to follow the improved working conditions. Using a practice sociology approach, with its emphasis on intersecting practice sets, we compared the accounts of workers with differing degrees of flexibility over working time and found only limited support for our proposition. The interview transcripts revealed a more nuanced picture of both flexibility and control than was elicited from the administered scales.

Standard notions of job control assume that the control refers to all aspects of the job, but our findings mirror survey studies which show that control at work is confined to specific aspects of the job and does not extend to the key elements associated with cardiovascular disease and impaired well-being that results from high job demands and low job control [23, 60]. Despite high autonomy across some aspects of work–setting start and stop times by professionals—our participants were unable to control employer demands which underpinned long hours, intense hours and security, and these pressures disrupted a sense of control over key elements of the job and of daily life as well as contributing to negative impacts on well-being and undermining approaches to staying healthy. The statutory provisions of Australia's Fair Work Act do little to address these fundamental demands on worker time.

The problems around working time globally have been extensively researched by the International Labour Organisation [61, 62]. Despite extensive legislation in advanced economies to regulate and improve choice over working hours the reality is that employee choices are often constrained by local labour market conditions, business operations and institutional support including transport, schooling, child care and health services. Evidence suggests that over employment (working longer than desired hours), under employment (working less than desired hours) and sequential mismatching around the hours of dual income families, especially linked to child care, bring additional scheduling challenges with travel and access to food and other services. With technological advances through the world-wide web and mobile phones, work can now be taken home, workers can be on call and work can be allocated outside of regulated work hours. The problems of work-time management have spread out from workplace health and safety to balancing work and family commitments and out-of-work health protection.

By foregrounding time as a key component to exhaustion and security, and by embedding working life within the temporal demands of other essential life domains—family, health, leisure—this study illustrates how the time strain-time adequacy dynamic is not only fuelled by

employment policies favouring relatively limited flexible work initiatives but by other social policies. It reveals the inadequacy of attending to work hours employment policies divorced from family-child care policy in tandem with gender equity policies, job security policies and health protection policies.

In terms of research directions, while Moen's theory of time strain incorporates the critically important element of "adequate time" to attend to personal and family matters, much could be gained by adapting the model of time strain- adequacy to more fully represent daily life realities. Under the existing model, time-adequacy combines factors within the job and outside of the job. Our research indicates that there is merit in developing a more extensive and stand-alone set of indicators of daily-life time adequacy, which would encapsulate life-stage characteristics such as caring responsibilities, as well as health conditions requiring long-term management, commute-to-work time and temporal capacity to meet evidence-based expectations regarding the preventive health practices of food consumption and routines, physical activity, leisure and sleep.

We also support recommendations [23] around the complementarity of working conditions surveys and in-depth observational studies, with the latter being most useful in eliciting over-looked factors. In the case of this study, the synergistic effects on job strain of the key performance movement alongside flexible work arrangements for professional workers has not to our knowledge been identified before.

Further in terms of future research and theory development, small qualitative studies are highly appropriate for investigating the social processes–socio-economic status identity formation, gender dynamics, transmission of cultural norms and practice routines–that can lie behind the often-hidden power struggles underway in different fields. A range of social processes emerged in this study as relevant to contestations around job control: the normalisation of long working hour employment regimes; acceptance of the individualisation of effort-reward in contrast to an earlier 'socialisation', or enterprise-level, effort-reward system; and related to this process, the acceptance of the nebulous metrics of performance as a new feature of job security. There are other more enduring sociological lines of enquiry: the perpetuation of the gendered nature of who the primary breadwinner and child carer is; and the acquisition of short-termist and flexible personal dispositions–and the dispensing of cultural rigidity—as a rationalisation for what some might see as risky approaches to health and well-being. Each of these processes could be the subject of in-depth exploration, yielding insights for action on the part of civil society, at least, as it tries to negotiate employment opportunities that contour around a meaningful social life rather than subordinating social life to working life.

## Supporting information

**S1 File. Employer recruitment letter.**
(DOCX)

**S2 File. Employee information letter.**
(DOCX)

**S3 File. Consent form.**
(DOCX)

**S4 File. Weekday time diary.**
(DOCX)

**S5 File. Sunday time diary.**
(DOCX)

**S6 File. Work time control protocol.**
(DOCX)

**S7 File. Interview protocol.**
(DOCX)

**S8 File. Background information + Job time strain.**
(DOCX)

## Author Contributions

**Conceptualization:** Jane Dixon, Lyndall Strazdins, John Burgess.

**Data curation:** Lara Corr.

**Formal analysis:** Lara Corr.

**Investigation:** Jane Dixon, Cathy Banwell, Lara Corr.

**Methodology:** Jane Dixon, Cathy Banwell.

**Writing – original draft:** Jane Dixon.

**Writing – review & editing:** Cathy Banwell, Lyndall Strazdins, Lara Corr, John Burgess.

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
