## [Decision Letter · Decision Letter 0]

6 Aug 2019

PONE-D-19-16021

Industrial relations transformed: temporal control and preventive health practices

PLOS ONE

Dear Dr Banwell,

Thank you for submitting your manuscript to PLOS ONE.  Two reviewers provided suggestions/comments on your manuscript and I also added some comments myself (see below).

After careful consideration, we feel that your manuscript has merit but does not fully meet PLOS ONE’s publication criteria as it currently stands. Therefore, we invite you to submit a revised version of the manuscript that addresses the points raised during the review process.

We would appreciate receiving your revised manuscript by Sep 14 2019 11:59PM. Please include the following items when submitting your revised manuscript:

We look forward to receiving your revised manuscript.

Kind regards,

Adrian Loerbroks

Academic Editor

PLOS ONE

 [No].

Please provide an amended Funding Statement that declares *all* the funding or sources of support received during this specific study (whether external or internal to your organization) as detailed online in our guide for authors at http://journals.plos.org/plosone/s/submit-now.  

Please state what role the funders took in the study.  If any authors received a salary from any of your funders, please state which authors and which funder. If the funders had no role, please state: "The funders had no role in study design, data collection and analysis, decision to publish, or preparation of the manuscript."

5. We note that your paper includes detailed descriptions of individual patients/participants. As per the PLOS ONE policy (http://journals.plos.org/plosone/s/submission-guidelines#loc-human-subjects-research) on papers that include identifying, or potentially identifying, information, the individual(s) or parent(s)/guardian(s) must be informed of the terms of the PLOS open-access (CC-BY) license and provide specific permission for publication of these details under the terms of this license. Please download the Consent Form for Publication in a PLOS Journal (http://journals.plos.org/plosone/s/file?id=8ce6/plos-consent-form-english.pdf). The signed consent form should not be submitted with the manuscript, but should be securely filed in the individual's case notes. Please amend the methods section and ethics statement of the manuscript to explicitly state that the patient/participant has provided consent for publication: “The individual in this manuscript has given written informed consent (as outlined in PLOS consent form) to publish these case details”.

6. Please consider changing the title so as to meet our title format requirement (https://journals.plos.org/plosone/s/submission-guidelines). In particular, the title should be "Specific, descriptive, concise, and comprehensible to readers outside the field" and in this case it is not informative and specific about your study's scope and methodology.

Additional Editor Comments:

Data availability: Please list a contact person or committee which can grant access to the data other than the corresponding author. According to Plos One guidelines "it is not acceptable for an author to be the sole named individual responsible for ensuring data access". (see https://journals.plos.org/plosone/s/data-availability

Methods:

- was the sample size determined a priori or is it the result of data collection until saturation?

- Did you use a topic guide? if so, please provide it

- Line 196-198: How does a presentation of findings contribute to validation? Did you discuss findings with the audience?

- The companies employed 60 and 200 workers, respectively. How exactly were participants selected from those samples? How many of the 28 individuals participated per company?

Results:

- do you have any additional descriptive data related to the participants (e.g. their age and type of contract, e.g. full time v spart time)?. Give your research questions it seems particularly interesting to what extent findings/themes differed among those with full-time vs part-time employment

- To what extent did you succeed in interviewing participants with a wide range of potentially relevant characteristics? E.g. did you manage to include all age groups, people with different employment schemes etc.? Please consider summarizing various of those descriptives in Table 1 (thus, you may want to expand that table)

Discussion: I would like to invite you to reflect in detail on how your methodological approach may have affected the nature of your findings (e.g. limitations); for instance, participants completed a diary on a single working day: to what extent can you be certain that this was a typical working day? One may suspect that dairies can only be completed on working days with higher than usual job control. Could this have affected the subsequent interviewing and thus the range of observations? This is just an example; in my view a much broader discussion of the potential limitations is required

Reviewer #1:

At the outset I must state that my expertise is in the sociology and anthropology of industrial labour and work, and hence not really into the debates and literature discussed in this paper. However, I very much enjoyed reading the paper. It presents a clear argument, and substantiates it with convincing evidence. Hence, I find the paper ready for being accepted for publication.

The only Thing that could be improved in my view is the theoretical debate the paper is pursuing. The authors of course relate their study to the work of Karasek, Moen, and Elchardus, but in my view they could further elaborate on the theses of the said authors and their critique of them. As it stands now they do so moe by passing than by seriously engaging with them. However, as indicated above, this is the opinion of a relative stranger to the field.

Reviewer #2

The authors discuss an important problem which is profoundly associated with the way in which we use theoretical concepts. This article has a specific point of view which is how “command over time” influences health behavior. However, the broader discussion regarding the development of “real” decision latitude in relation to formal structures and tightening markets is much older than the authors seem to be aware of. Despite that in the Scandinavian countries we have laws governing the right for workers to influence their own work situation several reviews based upon population surveys have revealed that broadly speaking perceived decision authority for working people has deteriorated during the past 35 years in various insidious ways. And similar discussions can be found already in Healthy Work (Karasek and Theorell 1990).The present authors discuss this in relation to health behavior but as stated above this discussion has been on stage for a long time with regard to more general aspects of work. See for instance Theorell T in: eds Cooper, Quick, Schabracq Handbook of work and Health Psychology 2009. The argument has been that in the modern world, “work flexibility” may be more beneficial for employers than for employees (similar arguments as in the authors´ text)!

I think the discussion needs to take in that this discussion has been on stage for a long time. The authors do mention such a discussion (Moen) but my argument is that it is a wider discussion.

With regard to the methodology. The qualitative approach that the authors have used serves as an eye opener. It could always be argued that the quotations may not necessarily be representative etc. And we are only talking about two work organizations in two branches. So more words of caution should be issued.

I feel that the examples presented are to a great extent focused on eating habits and physical activity. Health behavior is much more and it brings up the question whether the interviewers have had a narrow focus in their interviewing about health behavior.

But the discussion is interesting and brings up the important point that use of time must be analysed in a total context and that what may seem on surface as a good solution is not always so good.

---

## [Author Response · Author response to Decision Letter 0]

21 Sep 2019

Dear Dr. Loerbroks, 

Please find our point-by-point response to the matters raised by you and the 2 reviewers. We thank you all for valuable remarks and believe that we have strengthened the paper considerably as a result.

1. Please ensure that your manuscript meets PLOS ONE's style requirements

We have attended to the format instructions, but note that we need assistance with formatting of Tables 1 and 2. 

2. Please include additional information regarding the survey or questionnaire used in the study and ensure that you have provided sufficient details that others could replicate the analyses.

Pages 9-10 of the manuscript contains reference to the 5 time tool instruments and questionnaires that study participants responded to. Each has been uploaded as Appendices 4-8. 

3. We note that you have indicated that data from this study are available upon request. PLOS only allows data to be available upon request if there are legal or ethical restrictions on sharing data publicly. 

We are reluctant to release the interview transcripts into the public domain because there is a high likelihood given the small and select sample (N=28) that someone might be able to identify the respondents. Both work sites are located in regional towns and are small: one has 60 employees and the other 200. The transcripts contain respondents’ ages, education, gender, organisational role, family characteristics and other personal data. Given the descriptions that they provide of their workplaces within the transcripts, alongside the foregoing information, it would be relatively easy to identify each workplace and each respondent despite their de-identification. Furthermore as highlighted in our letter to you, we did not ask the participants for permission to share their transcripts. 

Please also provide contact information for a data access committee, ethics committee, or other institutional body to which data requests may be sent.

The study was approved by the Australian National University (ANU) Science and Medical Delegated Ethics Research Committee in 2014 (Approval 2014/285). ANU researchers are bound by the Australian Government’s Code for the Responsible Conduct of Research https://nhmrc.gov.au/research-policy/research-integrity/release-2018-australian-code-responsible-conductresearch.

An annual progress report of the research has been undertaken between 2014-2017 (when it was completed) in line with ANU Human Ethics procedures. The appropriate contact for this committee, now called Humanities & Social Sciences Delegated Ethics Review Committee, is: human.ethics.officer@anu.edu.au. See link to website

https://services.anu.edu.au/planning-governance/governance/humanities-social-sciences-delegated-ethics-review-committee

4. We have provided an amended Funding Statement that declares *all* the funding or sources of support received during this specific study (whether external or internal to your organization). 

a. Please state what role the funders took in the study. If any authors received a salary from any of your funders, please state which authors and which funder. If the funders had no role, please state: 

The study was funded by the Australian Research Council Discovery Programme grant number DP 140102856. The funders had no role in study design, data collection and analysis, decision to publish, or preparation of the manuscript. 

5. We note that your paper includes detailed descriptions of individual patients/participants… please provide a consent form following the PLOS guidelines.

All participants signed a Consent Form (uploaded as Appendix 3 alongside the Letter of Invitation-Information sent to employers and employees describing the study as Appendix 1 and 2) as required by the ANU Science and Medical Ethics Research Committee. We used the prescribed Committee form. All consent forms are kept in a secure location within the Research School of Population Health at the University, along with the transcripts and time tool forms completed by each participant. These materials will be kept for 5 years after the completion of the research (2017). It is now not realistic to return 4 years later to the 2 worksites and find the 28 individuals for the purposes of them signing the PLOS consent form. Moreover, it is perhaps unethical as they did not consent to be recontacted.

6. Please consider changing the title

The title has been changed from ‘’Industrial relations transformed: temporal control and preventive health practices’’ 

To “Flexible employment policies, temporal control and health promoting practices: A qualitative study in two Australian worksites”

Data availability: Please list a contact person or committee which can grant access to the data other than the corresponding author

Please see above under point 3. 

Methods: was the sample size determined a priori or is it the result of data collection until saturation? 

Lines 201-203 now read: We aimed to recruit a non-representative sample of around about 12 participants at each worksite based on research showing that new relevant information is rarely gained through additional interviews [45}

Did you use a topic guide? if so, please provide it 

Please see Appendix 8, now uploaded.

- Line 196-198: How does a presentation of findings contribute to validation? Did you discuss findings with the audience? 

Lines 228-232 now read: To validate the team’s interpretations, participating organisations received a summary of findings for their worksite and were provided with an oral presentation at a specially convened staff meeting. At the presentation, those interviewed and other interested staff could seek clarification or modification of interpretations. There was one substantial instance of the latter at the construction worksite.

- The companies employed 60 and 200 workers, respectively. How exactly were participants selected from those samples? How many of the 28 individuals participated per company?

The participants self-selected on the basis of management drawing attention to a flyer requesting volunteers sent to the workplaces by the study team. We sought a diverse range of employees and approximately 12 in each site. We were fortunate that 15 out of 200 employees in the larger worksite and 13 out of 60 in the small site responded. Table 1 on pages 11-12 sets out the characteristics of the 2 samples, and it indicates a range of job types, years with the employer and other indicators of a diverse pool of respondents in line with the methodological approach. 

Results: do you have any additional descriptive data related to the participants (e.g. their age and type of contract, e.g. full time v spart time)?. Give your research questions it seems particularly interesting to what extent findings/themes differed among those with full-time vs part-time employment

Table 1 contains the requested information. Only 2 of the 28 participants were part-time making any thematic differences with full-time workers elusive. We have however now included a new section (pages 25-28) entitled ‘’Common finding: Need for support from partners and extended family’’ which contains material as to why women may seek out part-time work.

… reflect in detail on how your methodological approach may have affected the nature of your findings (e.g. limitations); for instance, participants completed a diary on a single working day: to what extent can you be certain that this was a typical working day? One may suspect that dairies can only be completed on working days with higher than usual job control. Could this have affected the subsequent interviewing and thus the range of observations? 

We have included a new section Strengths and limitations (lines 739-759) at the end of the Discussion where we reflect on our methodological approach. As to the diary, participants were asked to complete one for a working day and one for a Sunday prior to, but close to, the interview. In the interviews, participants were asked how typical the entries were. They were very open about any discrepancies from a ‘normal’ day. 

Reviewer #1: The only Thing that could be improved in my view is the theoretical debate the paper is pursuing.

We have written a more fulsome Conclusion (an additional 470 words) to include discussion of the theoretical-research directions that we see as flowing from the study (see lines 781-831). Specifically, we propose that Moen’s conceptualisation is inadequate and that all studies of work demand-control have to consider the wider employment policy environment and societal conditions in which organisational-level dynamics are being operationalised. We note that the ongoing relevance of classic labour studies concerns.

Reviewer #2

the broader discussion regarding the development of “real” decision latitude in relation to formal structures and tightening markets is much older than the authors seem to be aware of

The original manuscript does contain a reference from 1989. However, we have now included 3 new references alongside the statement: “Job control insights have provoked decades of research activity and management practices [15,16,17,18]”. Two of these refer to Scandanavian research, as pointed to by the reviewer, as does reference 19.

… more words of [methodological] caution should be issued

Please refer to the new section on Study strengths and limitations

Health behavior is much more and it brings up the question whether the interviewers have had a narrow focus in their interviewing about health behavior. 

We are well aware of this point, but we intentionally chose to narrow the health risk factors to eating and physical activity for 2 reasons: they are major risk factors for obesity and weight related non-communicable disease and this is our area of expertise having written 2 books and numerous articles on the social determinants of obesity. Refer to lines 133-159 for a thorough review of the studies which focus on the possible links between flexible work and eating and physical activity. We did ask about sleep, as some research shows a link to weight, but did not include that analysis within the paper due to length blow-out. Some reference is made in the article to wanting more sleep as well as more leisure time. 

the discussion is interesting and brings up the important point that use of time must be analysed in a total context and that what may seem on surface as a good solution is not always so good.

We are pleased with this view, and have added some concluding remarks regarding employment policy directions on the basis of recent reports from EU bodies and the International Labour Organisation (refs 23, 60-62). The abstract also now has an additional paragraph reflecting this content. 

I am happy to follow up with any questions.

---

## [Decision Letter · Decision Letter 1]

17 Oct 2019

Flexible employment policies, temporal control and health promoting practices: A qualitative study in two Australian worksites

PONE-D-19-16021R1

Dear Dr. Dixon,

We are pleased to inform you that your manuscript has been judged scientifically suitable for publication and will be formally accepted for publication once it complies with all outstanding technical requirements.

With kind regards,

Adrian Loerbroks

Academic Editor

PLOS ONE

---

## [Editor Report · Acceptance letter]

12 Dec 2019

PONE-D-19-16021R1 

Flexible employment policies, temporal control and health promoting practices: A qualitative study in two Australian worksites 

Dear Dr. Dixon:

I am pleased to inform you that your manuscript has been deemed suitable for publication in PLOS ONE. Congratulations! Your manuscript is now with our production department. 

With kind regards,

on behalf of

Dr. Adrian Loerbroks 

Academic Editor

PLOS ONE